# Effect of IGFBP-4 during In Vitro Maturation on Developmental Competence of Bovine Cumulus Oocyte Complexes

**DOI:** 10.3390/ani14050673

**Published:** 2024-02-21

**Authors:** Adriana Raquel Camacho de Gutiérrez, Oguz Calisici, Christine Wrenzycki, Juan Carlos Gutiérrez-Añez, Christine Hoeflich, Andreas Hoeflich, Árpád Csaba Bajcsy, Marion Schmicke

**Affiliations:** 1Clinic for Cattle, University of Veterinary Medicine Hannover, Foundation, 30173 Hannover, Germany; adriff96@gmail.com (A.R.C.d.G.); calisicioguz@yahoo.com (O.C.); csaba.bajcsy@tiho-hannover.de (Á.C.B.); 2Clinic for Veterinary Obstetrics, Gynecology and Andrology of Large and Small Animals, Faculty of Veterinary Medicine, Justus-Liebig-University Giessen, 35392 Giessen, Germany; christine.wrenzycki@vetmed.uni-giessen.de; 3Medical-Surgical Department, College of Veterinary Medicine, University of Zulia, Maracaibo 4001, Venezuela; juan.gutierrez@fcv.luz.edu.ve; 4Ligandis UG, 18276 Gülzow-Prüzen, Germany; christine.hoeflich@ligandis.de; 5Institute of Genome Biology, Research Institute for Farm Animal Biology (FBN), 18196 Dummerstorf, Germany; hoeflich@fbn-dummerstorf.de

**Keywords:** IGFBP-4, IGF, oocyte, bovine embryos, in vitro embryo production

## Abstract

**Simple Summary:**

Insulin-like growth factors improve oocyte quality and are necessary for oocyte maturation. However, their effectiveness is contingent upon regulated availability, influenced by binding proteins. Specifically, IGFBP-4 has been identified as a factor blocking the positive effects of insulin-like growth factors, with higher expression observed in degenerating cattle follicles. Therefore, we studied the effect of IGFBP-4 on oocyte developmental potential using an in vitro embryo production system. We determined the ability of the oocyte to mature and cleave; proper embryo development; as well as the total cell number and some relevant genes for embryo quality. We found that IGFBP-4 did not affect the early stages of embryo development or growth. However, high IGFBP-4 concentrations decreased the number of embryos able to break and leave the Zona Pellucida, an important event before elongation and pregnancy establishment. We concluded that IGFBP-4 impairs embryo quality by decreasing the capacity of an embryo to hatch, and therefore the further development and pregnancy establishment could be impaired. These results provide new insights about the impact of IGFBP-4 on embryo quality and may have implications for cattle fertility.

**Abstract:**

Insulin-like growth factors (IGFs) are essential for oocyte maturation. Their bioavailability is regulated by their respective binding proteins (IGFBPs) and proteases. IGFBP-4 blocks the biological effects of IGFs. High IGFBP-4 expression has been associated with follicle atresia. We hypothesized that IGFBP-4 affects oocyte developmental competence during maturation. Therefore, the aim of this study was to examine the effect of IGFBP-4 on the developmental rate of bovine cumulus–oocyte complexes (COCs) during in vitro embryo production. Abattoir-derived COCs were matured with rbIGFBP-4 (2000, 540, and 54 ng/mL) compared to a control. Cumulus expansion, oocyte maturation, cleavage, blastocyst, and hatching rates were evaluated. Furthermore, blastocyst gene expression of SOCS2, STAT3, SLC2A1, SLCA3, BAX, and POU5F1 transcripts were quantified using RT-qPCR. No statistical differences were detected among the groups for cumulus expansion, maturation, cleavage, blastocyst rates, or all gene transcripts analyzed. However, at day 8 and 9, the number of total hatching and successfully hatched blastocysts was lower in 2000 ng/mL rbIGFBP-4 compared to the control (day 8: total hatching: 17.1 ± 0.21 vs. 31.2 ± 0.11%, *p* = 0.02 and hatched blastocyst 6.7 ± 0.31 vs. 21.5 ± 0.14%, *p* = 0.004; day 9 total hatching 36.4 ± 0.18 vs. 57.7 ± 0.10%, *p* = 0.009 and hatched blastocyst 18.2 ± 0.21 vs. 38.1 ± 0.11%, *p* = 0.004). We concluded that high concentrations of rbIGFBP-4 might negatively affect the subsequent ability of the embryo to hatch and possibly compromise further elongation.

## 1. Introduction

It is well known that the endocrine, autocrine, and paracrine actions of insulin-like growth factor 1 (IGF-1) and insulin-like growth factor 2 (IGF-2) affect oocyte maturation and subsequent developmental competence [1,2,3]. In bovine, IGF-2 and IGF-receptors (IGF-1R, IGF-2R) mRNA expression has been described in cumulus–oocyte complexes (COCs) and throughout the early embryonic stages. Their expression responded to temporal and spatial changes during COCs maturation and early embryo development [4,5,6]. During oocyte maturation, IGF-2 and their receptors increase. In contrast, during the early embryo stages, a subsequent decrease in IGF-2, IGF-1R, and IGF-2R transcription occurs following a gradual increase, reaching the highest levels up to hatching blastocyst stage [6]. 

In vitro maturation media supplementation with IGF-1, either alone or in combination with epidermal growth factor accelerates oocyte maturation and increases cumulus expansion and oocyte metabolism [7,8]. Moreover, IGF-1 in combination with angiotensin II improves oocyte developmental competence, increasing their subsequent blastocyst formation and hatching ability [9]. Similarly to IGF-1, IGF-2 enhances granulosa cell steroidogenesis and proliferation [2]. Both IGF-1 and IGF-2 (collectively referred to as IGFs) protect cumulus cells, oocytes, and embryos from apoptosis [10,11]. 

IGFs are bound to six high affine binding proteins (IGFBP1-6) [12]. The insulin-like growth factor binding proteins (IGFBPs) modulate IGFs actions, prolonging their half-life, serving as carrier proteins, and regulating their bioavailability at the target tissues [13]. It was shown that IGFBPs are also produced by the COCs [14]. In the oocyte microenvironment, IGFBPs are responsible for regulating the proliferative and antiapoptotic effects of IGFs on cumulus cells and oocytes [15,16]. Furthermore, IGFBPs have been proposed to modulate the agonistic effects of IGFs and gonadotropins (FSH and LH) on granulosa and theca cells [17]. Among the different binding proteins, IGFBP-4 is known as an inhibitor of the IGF action; mainly by sequestrating the IGFs, with a subsequent reduction in IGF bioavailability in the target tissues [12,18]. IGFBP-4 seems to be a critical factor in determining follicular fate, since IGFBP-4 levels decrease in dominant follicles compared to subordinate and atretic ones, mainly due to the acquirement of the proteolytic capacity for IGFBP-4 [19,20]. Furthermore, higher IGFBP-4 mRNA levels were detected in granulosa cells of atretic and persistent follicles [21], while lower IGFBP-4 mRNA expression was observed in the dominant follicles in cattle [22]. However, whether the IGFBP4 concentration in the follicular fluid also affects the oocyte directly or indirectly by sequestering IGFs has not been elucidated so far. In humans, IGFBP-4 exerts a positive effect on oocyte developmental competence [23]. Therefore, this study aimed to examine the effect of different IGFBP-4 concentrations on COCs during in vitro maturation and their subsequent embryo development. 

## 2. Materials and Methods

If not stated otherwise, all reagents and chemicals were purchased from Sigma-Aldrich (Taufkirchen, Germany).

### 2.1. Recombinant Bovine IGFBP-4

The recombinant bovine IGFBP-4 (rbIGFBP-4) was expressed by human embryonic kidney cells and in a concentration of 1.71 mg/mL, which was dissolved in 100 mM TRIS buffer with 300 mM NaCl, pH 8.5 (TRIS buffer, InVivo Biotech Services LLC; Hennigsdorf, Germany). The amino acid sequence used corresponded to the Uniprot accession number Q05716, amino acids 22-258. The original protein solution and stock solutions were filtered using a syringe filter (0.22 µm, Carl Roth, Karlsruhe, Germany) and stored in 50 µL aliquots at −20 °C for further use. The rbIGFBP-4 working solutions were prepared using our laboratory standard in vitro maturation (IVM). Three concentrations were tested based on previous studies showing stimulatory or inhibitory effects of IGFBP-4: high (2000 ng/mL) [24], intermediate (540 ng/mL) [25,26,27], and ten-times less as low (54 ng/mL) rbIGFBP-4, as final concentrations. Additionally, a medium control using IVM medium without rIGFBP-4 supplementation (0 ng/mL) and a sham control using IVM medium using TRIS buffer solution were included.

### 2.2. rbIGFBP4 Purity and Stability 

The purity and quantity of rbIGFBP-4 were verified by gel electrophoresis. About 8 µg rbIGFBP-4 was mixed with sample buffer (50 mM Tris, 2% SDS, 10% glycerol, 12.5 mM EDTA, 50 mM dithiothreitol, 0.0025% bromophenol blue; pH 6.8), boiled (1 min at 90 °C), and separated in a standard SDS Tris-base glycine polyacrylamide gel. A PageRuler™ pre-stained protein ladder (Thermo Fisher Scientific, Waltham, MA, USA) was used as a size marker. After separation, proteins in the gel were visualized by staining the gel with Colloidal Coomassie brilliant blue solution (0.02% Coomassie brilliant blue, 5% aluminum sulfate × 18 H_2_O, 10% ethanol, 2% orthophosphoric acid), which was then de-stained in water. The gel was imaged with use of a ChemiDoc Imaging System (Bio-Rad Lab., Hercules, CA, USA).

The rbIGFBP-4 stability was assessed in the working solutions with and without the presence of COCs. Previous to COCs placement into maturation dishes, three samples were taken. One sample was taken directly after the preparation of the working solution, one hour after being placed in the incubator, and directly before in vitro maturation started. From this time point onwards, samples were taken after 1, 3, 6, 12, and 24 h from the maturation dishes placed in an incubator at 38 °C and 5% CO_2_ in a humidified environment. Samples were examined using a qualitative Western ligand blotting, as previously described [28,29]. 

### 2.3. rbIGFBP4 Binding Capacity

The binding capacity of rbIGFBP-4 was assessed by supplementing IVM medium with rbIGFBP-4 (2000 ng/mL) and recombinant human IGF-2 at 100 ng/mL, as well as 50 ng/mL (rhIGF-2, 292-G2-050; R&D systems, Bio-techne, Abingdon, UK). The IGF-2 concentrations were chosen based on those in the literature previously described to exert positive effects on granulosa cells [2] and prevent oocyte degeneration [30]. Free IGF-2 was determined by a competitive radioimmunoassay (RIA) according to the manufacturer’s instructions (R30, Mediagnost^®^, Reutlingen, Germany). The pretreatment of samples using acidic buffer was omitted, so as to not alter the binding to rbIGFBP-4. IVM medium without rbIGFBP-4 or rhIGF-2 was used as control.

### 2.4. In Vitro Embryo Production

rbIGFBP-4 was used in 24 IVM runs to evaluate the expansion rate. All COCs of five IVM runs, were fixed and stained to analyze the maturation rate. The remaining 19 repetitions underwent IVF and were used for cleavage rate determination. The embryo culture system used in the present study allowed us to evaluate the embryo development up to 216 hours post insemination (hpi) without showing morphological degenerative changes. Thus, a total of 19 repetitions were used for evaluation of adequate blastocyst formation at 168 hpi. From these experiments, 18 repetitions were evaluated for late blastulation and early hatching at 192 hpi. Finally, 14 repetitions were evaluated for late hatching by calculating the hatching rate at 216 hpi.

#### 2.4.1. Recovery of Cumulus–Oocyte Complexes

COCs were collected by aspirating follicles from ovaries collected randomly from genitals without any apparent disease or pregnancy in two local abattoirs (Vion Bad Bramstedt LLC, Bad Bramstedt, Germany and Westfleisch SCE, Lübbecke, Germany) and were transported at 27 ± 3 °C in Dulbecco’s phosphate-buffered saline (PBS) containing 5.6 mg heparin (2 I.U.; SERVA electrophoresis LLC, Heidelberg, Germany) and 0.5 g bovine serum albumin (fraction V). Follicle aspiration was performed using an 18 G disposable needle (Easy-Lance, WDT, Garbsen, Germany) attached directly to a 12 mL disposable syringe (Henke-Ject^®^, Henke Sass Wolf, Tuttlingen, Germany). COCs were searched and classified using a stereomicroscope (SZX7, Olympus Corporation, Tokyo, Japan) within two consecutive hours following aspiration and were then placed into the IVM medium. Only COCs of grades I and II (homogeneous dark cytoplasm and at least three layers of compact cumulus cells) [31] were randomly assigned to the experimental groups. 

#### 2.4.2. In Vitro Maturation 

On the day of in vitro maturation, stock solutions of rbIGFBP-4 were thawed and 5 µL was added to 495 µL of IVM medium consisting of Earle’s salts HEPES modified tissue culture based medium 199 (TCM 199), which was supplemented with 1 mg/mL fatty-acid-free bovine serum albumin (FAF-BSA), 10 I.U./mL equine chorionic gonadotropin (eCG), and 5 I.U./mL human chorionic gonadotropin (hCG; Suigonan^®^ 80/40 I.U./mL lyophilizate and injection solution, MSD Animal Health, Unterschleissheim, Germany) and gently vortexed (Mini spin plus, Eppendorf, Hamburg, Germany) for 5 s. From the IVM medium, 250 µL was added to each well of a Nunc™ 4-Well culture dish (Nunclon^™^ Delta Surface, Thermo Fisher Scientific, Roskilde, Denmark). Dishes were equilibrated for 4 h in the incubator. Selected COCs were placed in groups with a mean of 22 and cultured for 23 to 24 h in vitro in a humidified environment at 38 °C under 5% CO_2_ (HERAcell^®^, Kendro Laboratory Products, Hanau, Germany). At the end of maturation, cumulus cell expansion was evaluated as “expanded” when all layers of cumulus cells had expanded except or including the corona radiata cells, or not “expanded” if minimal or no cumulus cells expansion was observed [32]. Expansion rate was estimated by dividing the number of expanded COCs by the total number of COCs having set in maturation.

#### 2.4.3. In Vitro Fertilization

For in vitro fertilization (IVF), Tyrodes albumin pyruvate medium (TALP) supplemented with 6 mg/mL FAF-BSA, 10 µM hypotaurine, 1 µM epinephrine, and 0.1 I.U./mL heparin (Heparin sodium, SERVA Electrophoresis LLC, Heidelberg, Germany) (Fert-TALP) solution was used.

In vitro fertilization was carried out using frozen-thawed semen from a bull with proven IVF fertility. Semen was thawed at 30 °C for one minute. Sperm separation was performed using a colloidal suspension for density gradient centrifugation (SpermFilter^®^ Gynotec B. V., Malden, The Netherlands). Thawed semen was carefully placed on a 750 µL TALP-SpermFilter^®^ solution (previously prepared by mixing 900 µL SpermFilter^®^ with 100 µL TALP solution) in a 1.5 mL Eppendorf^®^ tube and centrifuged for 16 min at 2400 rpm at room temperature (Mini Spin plus, Eppendorf, Hamburg, Germany). After centrifugation, the supernatant was removed, leaving 50 µL of the sperm pellet. This was washed with 750 µL TALP solution and centrifuged for three minutes. Then, the supernatant was removed, and a second washing step was performed using Fert-TALP solution with another three minutes of centrifugation. Finally, the supernatant was discharged, leaving 100 µL of purified sperm suspension. Sperm were counted in a Thoma cell counting chamber (Carl Roth, Karlsruhe, Germany) to calculate the insemination dose at a concentration of 1 × 10^6^ sperm/mL.

To perform the IVF, COCs were washed three times in 100 µL TALP solution drops and were placed into co-culture with the sperm in a 4-well dish containing 300 µL Fert-TALP for 18 to 20 h in a humidified 38 °C, 5% CO_2_ environment.

#### 2.4.4. In Vitro Culture

Cumulus cells of the presumptive zygotes were removed after 18–20 h of IVF by vortexing for five minutes. Remnant cumulus cells were loosened by gently pipetting to complete the removal of cumulus cells. Subsequently, presumptive zygotes were washed three times in 100 µL drops of in vitro culture medium consisting of synthetic oviductal fluid enriched with essential and non-essential amino acids (SOFaa) and incubated in 400 µL of IVC medium in 4-well dishes covered with 400 µL silicone oil (SERVA Electrophoresis, Heidelberg, Germany). The in vitro conditions consisted of a humidified atmosphere at 39 °C, with 5% CO_2_ and 5% O_2_ atmosphere for nine days.

Seventy-two hpi cleavage rate was calculated by dividing the total number of presumptive zygotes cleaved (≥2 blastomeres) by the total number of presumptive zygotes placed in culture.

Blastocysts rate was estimated at 168 and 192 hpi (days 7 and 8) by dividing the number of blastocysts by the number of presumptive zygotes set in culture. Hatching rate was measured at 192 and 216 hpi (day 8 and 9) by dividing the number of embryos hatching (zona broken, emerging of the zona and partially hatched blastocysts) by the number of total blastocysts developed. Hatched blastocyst rate was estimated by dividing the number of embryos successfully hatched (completely hatched zona free blastocysts) by the total number of blastocysts present. Total hatching blastocyst rate was calculated by dividing all blastocysts starting, partially, or completely hatched by the number of total blastocysts present. 

Finally, expanded, hatching and hatched blastocysts were washed and frozen in a PBS solution containing 0.1% PVA in a maximum volume of 5 µL and stored at –80 °C for further use.

### 2.5. mRNA Transcript Analysis

The mRNA expression of genes involved as IGFs mediators for cell proliferation and differentiation was analyzed. This included the signal transducer and activator of transcription 3 (STAT3) that has also been described as a regulator of trophoblast invasion capacity [33] and the suppressor of cytokine signaling 2 (SOCS2), an inhibitor of the IGF-induced intracellular JAK/STAT pathway [34,35]. Moreover, genes implicated in embryo metabolism such as the solute carrier family 2 member 1 (SLC2A1) [36] and the solute carrier family 2 member 3 (SLC2A3) [37], as well as the BCL2 associated X protein, the apoptosis regulator BAX (BAX) [10], and finally the gene related with embryo viability, POU class 5 homeobox 1 (POU5F1) [38] were selected as genes of interest. Detailed information about the primers used is given in Table 1.

#### 2.5.1. mRNA Isolation

To assess the effect of rbIGFBP-4 on the hatching process, single embryos showing a proper (day 8 hatching blastocyst) or delayed hatching time (day 9 hatching blastocyst) or unable to hatch (day 9 expanded blastocyst) were selected for mRNA expression. The mRNA was isolated using a Dynabeads™ mRNA DIRECT™ Micro Kit (Invitrogen by Thermo Fisher Scientific, Vilnius, Lithuania). The embryo lysis was performed using 150 µL lysis/binding buffer. Before RNA isolation, 1 pg rabbit globin mRNA (Sigma-Aldrich, Hamburg, Germany) was added to the samples as an exogenous reference gene. Samples were vortexed for 10 s and centrifuged at 13,000 rpm (Biofuge fresco, Heraeus, Kendro Laboratory Products, Osterode, Germany) for 10 min at room temperature (RT). For binding the poly (A)-chain of the embryonic mRNA to the Dynabeads, 10 µL of prewashed Dynabeads Oligo (dT)25 solution was added and incubated in a thermomixer at ~600 rpm (Thermomixer^®^ comfort, Eppendorf, Hamburg, Germany) for 5 min at RT. 

After incubation, the supernatant was removed using the MPC and washed once with 100 µL washing-buffer A (10 mM Tris-HCl, pH 7.5, 0.15 M LiCl, 1 mM EDTA, 0.1% LiDS) and three times using 100 µL washing buffer B (10 mM Tris-HCl, pH 7.5, 0.15 M LiCl, 1 mM EDTA). Finally, the mRNA was eluted using 11 µL of sterile water (Ampuwa^®^, KabiPac, Fresenius Kabi, Bad Homburg, Germany) at 65 °C (Thermomixer^®^ FA, Eppendorf, Hamburg, Germany) for 3 min. The MPC was placed into ice, and the supernatant was collected and maintained on ice for use in reverse transcription (RT-PCR).

#### 2.5.2. Reverse Transcription

For the reverse transcription, a reaction mix was prepared with a final concentration of 2.5 µM random hexamers (Invitrogen by Thermo Fisher Scientific, Vilnius, Lithuania), 1 mM of deoxyribonucleotide triphosphate (dNTPs) nucleotides (Eurogentec, Cologne, Germany), and 5 mM of MgCl2 (Invitrogen by Thermo Fisher Scientific, Vilnius, Lithuania) in 1 x PCR buffer (50 mM KCl, 20 mM Tris-HCl, 5 mM MgCl2 pH 8.4; Invitrogen by Thermo Fisher Scientific, Vilnius, Lithuania). This was completed by 50 I.U. murine leukemia virus (MuLV) reverse transcriptase (Invitrogen by Thermo Fisher Scientific, Vilnius, Lithuania) and 20 I.U. RNase-Inhibitor (Applied Biosystems by Thermo Fisher Scientific, Schwerte, Germany). Then, 11 µL of isolated mRNA was added to the mastermix solution. Finally, Ampuwa^®^ water (KabiPac, Fresenius Kabi, Bad Homburg, Germany) was added to reach 20 µL total reaction volume.

Reverse transcription was performed for 10 min at 25 °C and for 60 min at 42 °C, followed by a denaturation step for 5 min at 99 °C. Finally, the samples were kept on ice and were immediately used for quantitative PCR (q-PCR) analysis.

#### 2.5.3. Quantitative Polymerase Chain Reaction (qPCR)

qPCR was performed using a PCR reaction mix by adding 10 µL MESA GREEN qPCR MasterMix Plus for SYBR^®^ Assay w/ fluorescein reaction buffer mixture, containing the deoxyribonucleoside triphosphate (dNTPs), Meteor Taq DNA polymerase, MgCl_2_, SYRB^®^ Green I, stabilizers, fluorescein (Eurogentec, Cologne, Germany), 0.2 µM forward primer, and 0.2 µM of the reverse primer (MWG Eurofins Genomics, Ebersberg, Germany; Table 1). cDNA concentration was calculated for globin and for each gene of interest as an embryo equivalent, varying from 0.25 up to 2 per blastocyst. Finally, Ampuwa^®^ water was added to reach a total reaction volume of 20 µL.

For each primer pair, all reactions were run in duplicate. Three negative controls were used (nuclease-free water, second negative control containing RNA but no RNAse inhibitor and no reverse transcriptase, and a third negative control containing RNAse inhibitor and reverse transcriptase but no RNA). A standard curve consisting of the globin cDNA dilution containing 50, 25, 12.50, and 6.25 fg RNA was used.

The mRNA was measured using a BioRad Real-Time System CFX96 1000 Touch (BioRad, Munich, Germany). The cycling program first consisted of a protein denaturation step at 95 °C for 10 min. For the amplification phase, a total of 43 cycles were performed as follows: 95 °C for 15 s for denaturation, 60 °C for 30 s for annealing, and 72 °C for 30 s for elongation. The dissociation was performed at 95 °C for 15 s. Finally, a melting curve with 0.5 °C steps every 10 s from 55 °C up to 95 °C was used to verify the identity of the PCR fragments.

The relative abundance of gene transcripts of at least four biological replicates obtained from different IVF cycles was calculated according to Kuzmany et al. (2011) [39], through a Delta-delta-Ct approach using the equation 4 proposed by Pfaffl (2001) [40]. The relative abundance was calculated using the globin efficiency for the primer pair in each run. The efficiency range between the target and the endogenous control amplifications was set between 80 and 100%.

### 2.6. Assessment of Nuclear Maturation by Aceto-Orcein Staining

After maturation, COCs were denuded by incubation with hyaluronidase solution for 10 min at 37 °C and then vortexed for 5 min. COCs were fixed in an ethanol-acetic acid solution (three parts ethanol and one part acetic acid) for 24 h and stained with 1% orcein solution (1 g orcein in 100 mL ethanol-acetic acid solution). Nuclear maturation stage was evaluated under a phase contrast microscope (Axiostar plus, Carl Zeiss, Jena, Germany) using a contrast phase filter N° 2 at 400× magnification. Oocytes were classified as germinal vesicle (GV), germinal vesicle breakdown (GVBD), metaphase I (MI), or metaphase II (MII). Nuclear maturation was calculated by the number of oocytes that achieved metaphase II (MII) stage divided by the total number of evaluated oocytes.

### 2.7. Statistical Analysis

A normality test was performed with a Shapiro–Wilk W test and unequal variance through Levene test. IVF outcome data followed a non-normal distribution. 

The categorical data from IVF outcomes and embryo developmental rates (e.g., expansion, oocyte maturation, cleavage, blastocyst, and hatching rates) were compared using a chi-square test (SAS^®^ Version 9.4, Cary, NC, USA). Similarly, odds ratio estimates and confidence interval limits for relative risks of embryo developmental rates were calculated using the logistic procedure (PRO LOGISTIC) from the same statistical package. 

On the other hand, the relative expression of gene transcripts was analyzed using a generalized linear model (GLM) from JMP (SAS1). First, a normality test was performed with a Shapiro-Wilk W test, while unequal variance was evaluated through a Levene test. The means difference among the treatments was evaluated through Tukey–Kramer test for normally distributed data. For non-normal distributed data, a non-parametric approach through a Steel–Dwass test was performed. A significant difference was defined at a *p* value < 0.05.

## 3. Results

### 3.1. rbIGFBP-4 Purity and Stability

rbIGFBP-4 could be confirmed intact as a single band corresponding to 25 kDa in the specified concentration (1.71 mg/mL; Figure 1).

### 3.2. rbIGFBP4 Binding Capacity 

It could be proven that the used rbIGFBP-4 was able to bind biotinylated IGF-2. rbIGFBP-4 was detected before and after 1, 3, 6, 12, and 24 h during incubation at 38 °C in the presence or absence of COCs, reflecting the stability of the protein. No rbIGFBP-4 was detected in the negative medium control or the negative control with the presence of COCs (Figure 1). 

Moreover, using RIA, a mean of 15.6 ± 2.9% rhIGF-2 added in IVM medium supplemented with 2000 ng/mL of rbIGFBP-4 was detected, which means that 84.4 ± 2.8% of rhIGF-2 was bound to rbIGFBP-4. No rhIGF-2 was detected in the medium control.

### 3.3. Developmental Rates of COCs and Embryos

No statistically significant differences were detected between the rbIGFBP-4 groups and medium control regarding the expansion rate, maturation (Figure 2), cleavage, or blastocyst rate at 168, 192, and 216 hpi (Table 2). 

Total hatching rate at 192 hpi (day 8) was higher in the medium control than with 2000 ng/mL rbIGFBP-4 (Figure 3). The estimated odds ratio was 2.25, with a confidence interval (CI) ranging between 1.175 and 4.314. Moreover, in medium with 2000 ng/mL rbIGFBP-4, a lower number of hatched blastocysts at day 8 was detected in comparison with the control group (Figure 3). The control group showed an odds ratio 3.83 times greater for completing the hatching process (CI: 1.566–9.375) in comparison to 2000 ng/mL rbIGFBP-4. 

Similarly, at 216 hpi (day 9), the total hatching and hatched blastocyst rate was higher in the medium control than 2000 ng/mL rbIGFBP-4 (Figure 3). The probability of hatching occurrence for the medium control group was 70% greater (odds ratio: 2.39, CI: 1.346–4.245) than 2000 ng/mL rbIGFBP-4.

### 3.4. Relative Abundance of mRNA Transcripts

The transcripts levels of SOCS2, STAT3, SLC2A1, SLC2A3, BAX, and POU5F1 were similar among the experimental groups for each embryo category and developmental day (Figure 4).

## 4. Discussion

This study marks the first investigation of the impact of adding rbIGFBP-4 during in vitro maturation. 

Previously to IVM, we could confirm rbIGFBP-4 stability and a high binding capacity of 2000 ng/mL rbIGFBP-4 to IGF-2. Bovine and human IGF-2 show a high similarity [41,42]. Notably, rbIGFBP-4 demonstrated its biological function by sequestering more than 80% of human IGF-2 using 2000 ng/mL concentration. Previous studies described that 20 nM (approximately 540 ng/mL) blocked the effects of 2 nM IGF-1 (around 15 ng/mL) in human trophoblast cells [25]. Additionally, 20 nM, which is double equimolar to 100 ng/mL IGF-1, inhibited proliferation in rat epithelial cells [26]. Moreover, exogenous IGFBP-4 (500 ng/mL) also inhibited proliferation but enhanced the differentiation of mice neural progenitor cells [27]. Other in vitro studies found that high exogenous IGFBP-4 (1000 to 2000 ng/mL) exerted proliferation and differentiation in cardiomyocytes derived from murine-induced pluripotent stem cells [24]. Therefore, a defined range of rbIGFBP-4 concentrations were tested in the present study. However, in future studies a finer concentration gradient could be tested.

We found that rbIGFBP-4 neither affected the percentage of MII-stage oocytes nor cumulus expansion 22–24 h post IVM and did also not affect embryo production until 192 hpi. Similar results were found by Giroto et al. (2018) [43], as they added IGFBP-4 proteolytic enzyme PAPP-A during the COCs IVM, suggesting that the autocrine and paracrine effect of IGFs produced by COCs during IVM did not affect the capacity of the oocytes for blastocyst formation. Moreover, Nicholas et al. (2005) [44] did not find any differences in the cleavage rate of IVM oocytes derived from follicles with different IGFBP-4 profiles. In contrast to our study, they described a better blastocyst development after 7 days of culture within the COCs derived from follicles with higher IGFBP-4 profiles. Nevertheless, in their study, high concentrations of IGFBP-4 were found in the early atretic follicular stage.

In humans, Wang et al. (2006) [23] showed that COCs derived from different concentrations of IGFBP-4 in follicular fluid prior to IVM exerted different effects on oocyte developmental capacity. In their study, the lowest IGFBP-4 concentration was detrimental to embryo development, and better embryo development was found in COCs derived from the highest IGFBP-4 concentrations in follicular fluid. Nonetheless, the highest concentration referred to 90.2 ± 39.0 ng/mL, which nearly enclosed the lowest concentration used in the present study, thus indicating that IGFBP-4 could also act on the oocytes during the final stage of follicle development. Unfortunately, bovine follicular fluid IGFBP-4 concentrations are not available in the literature. However, in heifers, after two synchronized embryo transfer cycles, the maternal serum concentration of IGFBP-4 was approximately 500 ng/mL in pregnant and near to 900 ng/mL in non-pregnant heifers [45]. Furthermore, it was shown that cows with high serum concentrations of IGFBP-4 were associated with pregnancy loss within the first 60 days after artificial insemination [46]. In another study from our working group, similar results were found in lactating cows with pregnancy loss, where IGFBP-4 serum concentrations were as high as 2000 ng/mL (unpublished data).

Previous studies described that the activation of the JAK/STAT signaling pathway modulatesembryo-maternal secreted factors that increase hatching [47] Moreover, the glucose transporters SLC2A1, SLC2A3 are essential for the energetic metabolism of embryos during blastulation and hatching [48,49]. In our study, no effect of IGFBP-4 on the evaluated blastocyst categories was found. Early studies have shown the effect of IGFs on the gene expression related with vital biological functions of bovine COCs and preimplantation embryos, such as BAX, IGF-1R, IGFBP-3, SLC2A8, heat shock protein 70, desmocollin II, and Na/K-ATPase [10,11,50,51,52,53]. In contrast, in the present study, IGFBP-4 neither affected the previously described IGFs-modulated gene expression of SOCS2 nor STAT3 for cell differentiation and proliferation nor any genes involved in oocyte metabolism. This led us to the hypothesis that IGFBP-4 might exert an IGFs-independent effect on COCs. IGFBP-4 evoked IGFs-independent effects by inhibiting the canonical WNT/ꞵ-catenin signaling pathway [54,55]. That pathway plays an essential role in oocyte developmental competence acquisition during IVM and embryo development [56,57]. This might be an intriguing cell signaling transduction pathway for future investigations. Some of the tested factors are intracellular stored factors in which we do not necessarily see differences at the gene level. A limitation of this study was that we did not have enough material of the single embryos to test all factors in a Western blot at protein level. Nevertheless, further evaluations of the differences in the protein levels of the study genes and other components from the cellular pathways evaluated in the present study are required to confirm these hypotheses.

Interestingly, the number of blastocysts where hatching was initiated and successfully carried out (completed out of the zona pellucida) was negatively affected by the highest concentration of IGFBP-4 but not affected by the medium and low IGFBP-4 concentrations. These results demonstrate that IGFBP-4 has a concentration dependent effect on COCs and thus can be expressed in the hatchability of embryos. Previous in vitro studies showed a dose-dependent effect of exogenous IGFBP-4 with other tissues culture such as cardiomyocytes [24] and bone cells [58]. These results could be explained by the inhibition of the positive effect of IGF-2 on cell proliferation and differentiation, expressed as an increase in the total cell number in bovine embryos [5,59]. Embryos with higher cell numbers and less apoptotic cell ratios expressed higher hatching ability [60,61,62]. A critical point in this study is that the concentrations of IGF-1 or IGF-2 were not both assessed. Therefore, further studies are needed to clarify if the tested effect might be a direct IGFBP-4 effect or an indirect effect from sequestering IGFs. Therefore, in further studies, the concentration of free IGFs should also be carefully determined. 

Moreover, enzymatic degradation of the zona pellucida through certain proteases, specifically in bovine urokinase-type plasminogen activator, was described to be involved in the hatching process [63,64]. Several studies have described an interaction between the plasminogen activators/plasminogen inhibitors system and the IGF system [65]. The plasminogen system not only participates in the blastocyst hatching process but also in follicular development, oocyte maturation, and ovulation [66,67]. Moreover, it has been described that a regulatory loop exists between the IGFBP-4/IGFs and the plasminogen activator/plasminogen inhibitor system [65,68]. In rabbit, ovarian perfusion using IGF-1 showed an increase in intrafollicular plasminogen activator activity [69]. Whether IGFBP-4 might affect the intrinsic capacity of COCs for urokinase-type plasminogen activator production in an IGFs dependent way, thus affecting their posterior hatching ability, is still unknown. These preliminary results open a wide range of questions and warrant further investigations.

## 5. Conclusions

Although the precise mechanism by which rbIGFBP-4 affects the hatching ability of embryos derived from COCs matured under high concentrations of rbIGFBP-4 could not be described in all its details, our study provides the first insight into an rbIGFBP-4 effect on COCs during IVM. We conclude that rbIGFBP-4 has a detrimental effect on the embryo hatching ability of embryos derived from COCs matured in vitro with IGFBP-4 in a dose-dependent manner.

## Figures and Tables

**Figure 1 animals-14-00673-f001:**
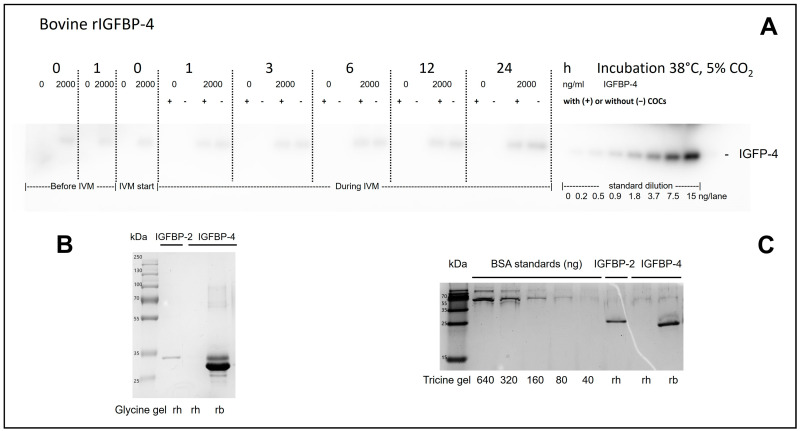
(**A**) Analysis of in vitro maturation (IVM) medium with and without addition of recombinant bovine insulin-like growth factor protein 4 (rbIGFBP-4). rbIGFBP-4 could be detected as a band in all samples in which rbIGFBP-4 (2000 ng/mL) was added, prior to the start of IVM, samples were taken; (0) directly after IVM medium preparation, (1) one hour after incubation. Prior to IVM (0), a sample was taken before cumulus–oocyte complexes (COCs) were placed in the maturation plates. During IVM, samples were taken from wells in the presence (+) or absence (− of COCs 1, 3, 6, 12, and 24 h after IVM started. The Western ligand blot image has been cropped at the level corresponding to IGFBP-4, an original image of the gel is provided in the Appendix A. (**B**) analysis of intact recombinant human IGFBP-4 (rhIGFBP-4), rbIGFBP-4, and recombinant human (IGFBP-2). (**C**) Protein quantity determination of rh- and rbIGFBP-4 and rhIGFBP-2.

**Figure 2 animals-14-00673-f002:**
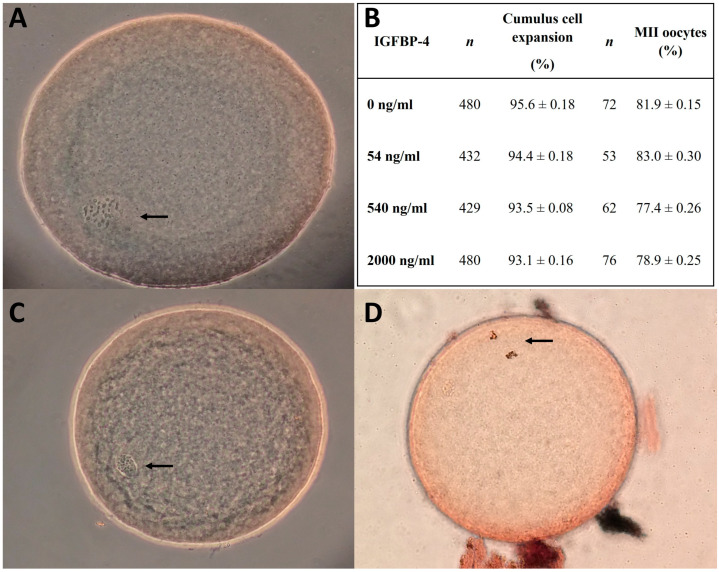
Cumulus expansion and nuclear maturation stage in oocytes, as a sign of successful oocytes in vitro maturation (IVM) using aceto-orcein staining. (**A**) Representative image of an oocyte in germinal vesicle breakdown (GVBD; indicated by the arrow) 22–24 h after IVM. (**B**) Cumulus cell expansion and metaphase II (MII) oocyte rate after 22–24 h IVM. The data represent means ± SEM (24 replicates and 4 replicates, respectively). No statistical difference was found among the treatment groups. Data were analyzed using an Xi square test with *p* ˃ 0.05. (**C**) Representative image of an oocyte in germinal vesicle stage (GV; indicated by the arrow) 22–24 h after IVM. (**D**) Representative image of an oocyte in MII stage 22–24 h after IVM (indicated by the arrow). Images assessed by phase contrast microscopy under 400× magnification.

**Figure 3 animals-14-00673-f003:**
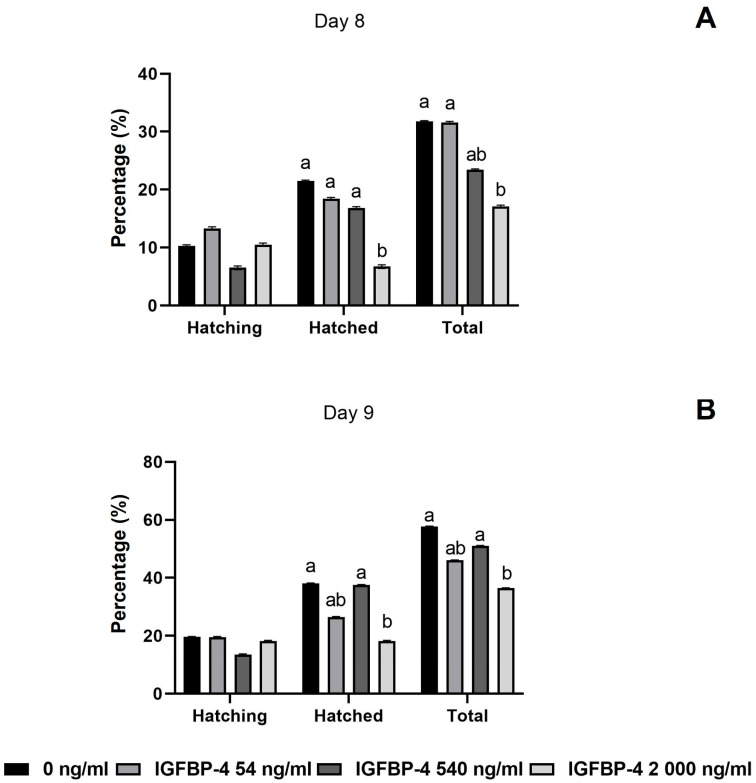
Effect on the hatching capacity of blastocysts at (**A**) day 8 (192 hpi) and (**B**) day 9 (216 hpi) derived from COCs matured with/without IGFBP-4. Different superscripts within each variable represent statistically significant differences between the experimental groups (a:b; *p* < 0.05).

**Figure 4 animals-14-00673-f004:**
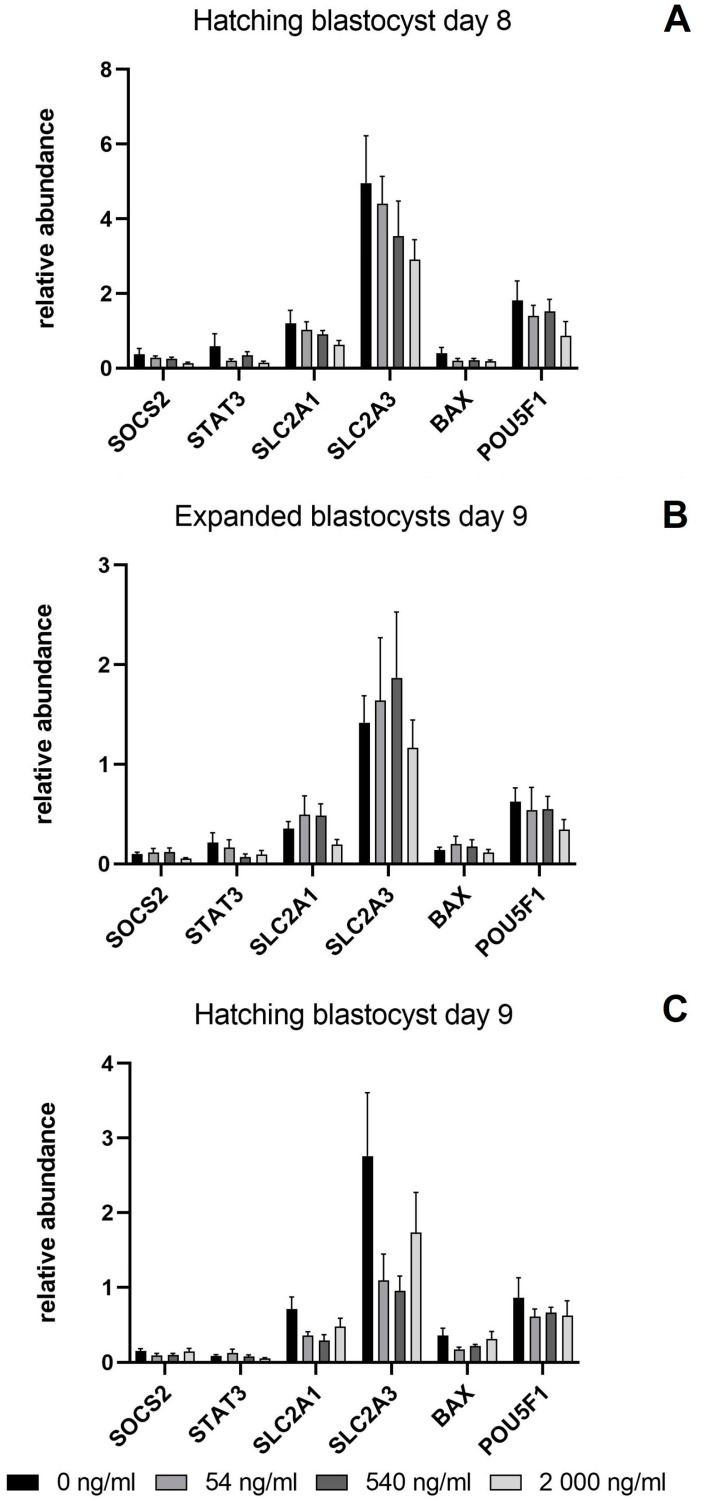
Relative abundance of the transcripts of the genes of interest. (**A**) mRNA expression from bovine hatching blastocyst day 8 (192 hpi) derived from COCs in vitro matured with or without IGFBP-4. (**B**) mRNA expression from bovine expanded blastocyst day 9 (216 hpi) derived from COCs in vitro matured with or without IGFBP-4 (**C**) mRNA expression from bovine hatching blastocyst day 9 (216 hpi) derived from COCs in vitro matured with and without IGFBP-4 derived of COCs matured with or without IGFBP-4. Values are shown as means ± S.E.M.

**Table 1 animals-14-00673-t001:** Genes of interests used for qPCR.

Gene	Strand	Primer Sequenz 5′-3′	Amplicon	Accession Number
			Length (bp)	
*Globin*	forward	GCA GCC ACG GTG GCG AGT AT	257	X04751
	reverse	GTG GGA CAG GAG CTT GAA AT		
*SOCS2*	forward	GTG TGG CAA GGT AGC TAG GG	202	NM_177523.2
	reverse	TAC CAG TGC TGG GAC CTT TC		
*STAT3*	forward	TCT ACC CCG ACA TTC CAA AG	157	NM_001012671
	reverse	GGC AGG TCA ATG GTA TTG CT		
*SLC2A1*	forward	CAG GAG ATG AAG GAG GAG AGC	258	NM_174602
	reverse	CAC AAA TAG CGA CAC GAC AGT		
*SLC2A3*	forward	ATC CCT GTG GTC CTT GTC TG	202	NM_174603
	reverse	GAT AAT CAG TCG GCC CAA GA		
*BAX*	forward	TCT GCA GGC AAC TTC AAC TG	199	NM_173894
	reverse	TGG GTG TCC CAA AGT AGG AG		
*POU5F1*	forward	GAG GAG TCC CAG GAC ATC AA	149	NM_174580
	reverse	GTC GTT TGG CTG AAC ACC TT		

bp = base pairs.

**Table 2 animals-14-00673-t002:** The effect of rbIGFBP-4 during in vitro maturation on oocyte developmental competence of adult bovine oocytes derived from abattoir ovaries. Percentages of cumulus oocyte expansion, presumptive zygotes cleaved, and blastocyst formation (mean ± SEM).

				Blastocyst (%)				
rbIGFBP-4	*n*1	Cleavage (%)	*n*2	168 hpi	*n*3	192 hpi	*n*4	216 hpi
**0 ng/mL**	450	71.6 ± 0.09	400	21.0 ± 0.10	408	26.2 ± 0.09	334	29.0 ± 0.10
**54 ng/mL**	412	71.4 ± 0.09	412	18.7 ± 0.10	371	26.4 ± 0.10	287	30.3 ± 0.11
**540 ng/mL**	414	70.8 ± 0.05	414	21.3 ± 0.06	373	28.7 ± 0.05	288	33.3 ± 0.06
**2000 ng/mL**	430	69.5 ± 0.09	386	15.8 ± 0.11	388	27.1 ± 0.09	304	32.6 ± 0.10

*n:* Number of oocytes that underwent in vitro fertilization, number of oocytes that underwent in vitro culture until 168, 192, and 216 h post insemination. For expansion rate, 24 replicates were performed. From these, 19 replicates underwent IVF and were used for cleavage and blastocyst rate 168 hpi; 18 replicates were evaluated until 192 hpi; and finally, 14 repetitions were evaluated until 216 hpi.

## Data Availability

The data presented in this study are available on request from the corresponding author. The data are not publicly available due to arrangement and structure of the data.

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
