# Peer review of "Effect of IGFBP-4 during In Vitro Maturation on Developmental Competence of Bovine Cumulus Oocyte Complexes"

_animals, 2024, doi:10.3390/ani14050673_

Round 1

Reviewer 1 Report

Comments and Suggestions for Authors

The article is well written and prepared, it is relevant and interesting to the scientific and technical community, however there are some points that are not very clear to the reader and others that need to be revised and improved.

The experimental design needs to be better explained, how many oocytes each repetition started and from these oocytes what interventions were made?

I understand that cleavage evaluation was carried out and then the same structures were evaluated at 168, 192 and 216 hpi. At these times, how many embryos were removed from the treatment groups for gene expression analyses? Were 19x, 18x and 14x repetitions for moments 168, 192 and 216 hpi performed independently or concomitantly? Why is there a difference in the number of repetitions for the 7, 8 and 9 day? I suggest explaining the methodology better.

In item 2.4.1. Recovery of cumulus-oocyte complexes.

Were the COCs collected at the slaughterhouse or just the ovaries and then transported to the laboratory where the follicles were aspirated? Is the transport temperature of 27°C correct?

In item 2.4.3. In vitro fertilization.

Is the temperature used to thaw semen correct?

Table 2. The group with the best result is the one with the highest blastocyst rate. At 192 and 216 hpi, the group treated with 540 ng/mL was the one that obtained the highest blastocyst (%), but the discussion is different. Check the interpretation or the way in which the results were presented.

The results of gene expression analysis have been little explored, I suggest correlating them with observations found in embryonic development.

In the discussion there is the statement: “Notably, more than 80% of IGF-2 was sequestered with this concentration”. Where is this result? Show based on the results observed and reference it in the discussion, highlighting this statement.

Based on these adjustments, adapt the presentation of results and discussion.

Author Response

Dear Review,

Thank you for your comments. Please see the attachment.

Best regards,

Adriana Camacho de Gutiérrez

Reviewer 2 Report

Comments and Suggestions for Authors

This paper examined the effect of IGFBP-4 on developmental rate of bovine cumulus-oocyte complexes (COCs) during in vitro embryo production by treatment of rbIGFBP-4 and blastocyst gene expression, and found rbIGFBP-4 had a detrimental effect on the embryo hatching ability of embryos derived from COCs matured in vitro with IGFBP-4 in a dose-dependent manner. It is meaningful to improve embryo quality.

This manuscript is well written, with interesting results that is supported by relevant research. Some problems in this paper lacked sufficient data to prove the results:

1. Gene names need to be defined at first use in the manuscript.

2. Line 95-100: Why 54 ng/ml, 540ng/ml, 2000ng/ml were chosen as concentration gradients for IGFBP-4? Please add details.

3. Line 120, in “rbIGFBP4 binding capacity”, the binding capacity of rbIGFBP-4 was accessed by supplementing IVM medium with rbIGFBP-4 (2,000 ng/ml) and recombinant human IGF-2 at 100 ng/ml as well as 50ng/ml. Hear, “human” IGF-2 was used, why not “bovine” IGF-2? The detail information needed.

4. Line 135-136: Why these three times were chosen for the study? The detail information needed.

5. Line 138: More details are needed for the collection of COC. How are the animals chosen? Just random sampling? Please provide the animal information of the COC samples, including parity, age, days after calving, etc.”

6. Line 224: The second column name of the table is missing. Please be more specific.

7. Line 227: What was the average RNA integrity number?

8. Line 303 and Line 308: This section was very hard to follow. Suggest the authors add details of the maximum likelihood estimation analysis and the generalized linear model and clarify the meaning of each parameter in the models.

9.  Line 332: Please use n1, n2, n3, and n4 to represent number for oocytes in different stages, respectively.

10. Line 349, in “3.4. Relative abundance of mRNA transcripts”, only transcripts levels of SOCS2, STAT3, SLC2A1, SLC2A3, BAX, and POU5F1 were detected for each embryo category and developmental day, but the protein levels of these 6 genes was not clear, it should be added.

11.  Line 350-352: There is only one sentence description of this part, please add more details in revised manuscript.

12.  Line 354: Please reposition "A" and harmonize the font size in Figure 1.

13.   Line 375: What does the y-axis represent? Please modified Figure 3.

14. Line 425: “Early studies have shown the effect of IGFs on the gene expression related with vital biological functions of COCs and preimplantation embryos”. What species do these studies focus on? Please add details

15. Please do a thorough read through to ensure all grammatical errors are corrected.
